# Association between Time Spent on Smart Devices and Change in Refractive Error: A 1-Year Prospective Observational Study among Hong Kong Children and Adolescents

**DOI:** 10.3390/ijerph17238923

**Published:** 2020-11-30

**Authors:** Chi-wai Do, Lily Y. L. Chan, Andy C. Y. Tse, Teris Cheung, Billy C. L. So, Wing Chun Tang, W. Y. Yu, Geoffrey C. H. Chu, Grace P. Y. Szeto, Regina L. T. Lee, Paul H. Lee

**Affiliations:** 1Centre for Myopia Research, School of Optometry, Hong Kong Polytechnic University, Hong Kong, China; chi-wai.do@polyu.edu.hk (C.-w.D.); lily.yl.chan@polyu.edu.hk (L.Y.L.C.); wing.tang@polyu.edu.hk (W.C.T.); lydia.yu@polyu.edu.hk (W.Y.Y.); geoffrey.chu@polyu.edu.hk (G.C.H.C.); 2Department of Health and Physical Education, Education University of Hong Kong, Hong Kong, China; andytcy@eduhk.hk; 3School of Nursing, Hong Kong Polytechnic University, Hong Kong, China; teris.cheung@polyu.edu.hk; 4Department of Rehabilitation Sciences, Hong Kong Polytechnic University, Hong Kong, China; billy.so@polyu.edu.hk; 5School of Medical and Health Sciences, Tung Wah College, Hong Kong, China; grace.szeto@polyu.edu.hk; 6School of Nursing and Midwifery, University of Newcastle, Callaghan, NSW 2308, Australia; Regina.L.Lee@newcastle.edu.au

**Keywords:** handheld device, myopia, prospective, smartphone, tablet, teenage

## Abstract

This study examined the association between smart device usage and the 1-year change in refractive error among a representative sample of Hong Kong children and adolescents aged 8–14 years. A total of 1597 participants (49.9% male, mean age 10.9, SD 2.0) who completed both baseline (2017–2018) and 1-year follow-up (2018–2019) eye examinations were included in the present study. The non-cycloplegic auto-refractive error was measured and the average spherical equivalent refraction (SER) was analyzed. The participants also self-reported their smart device usage at baseline. Multivariate regression adjusted for age, sex, baseline SER, parents’ short-sightedness, BMI, time spent on moderate-to-vigorous physical activity (MVPA), and caregiver-reported socio-economic status showed that, compared with the reference group (<2 h per day on both smartphone and tablet usages), those who spent ≥2 h per day using a smartphone and <2 h per day using a tablet had a significantly negative shift in refractive error (1-year change in SER −0.25 vs. −0.09 D, *p* = 0.01) for the right eye, while the level of significance was marginal (1-year change −0.28 vs. −0.15 D, *p* = 0.055) for the left eye. To conclude, our data suggested spending at most 2 h per day on both smartphones and tablets.

## 1. Introduction

Uncorrected refractive error is the leading cause of visual impairment worldwide, according to reports by the World Health Organization [1]. Short-sightedness, also referred to as myopia, is the most common cause of uncorrected refractive error [2]. The progression of myopia has been a concerning topic from a public health perspective due to the increased risks it poses for numerous visually blinding conditions such as glaucoma and retinal detachment [3,4,5]. Myopia is becoming an epidemic worldwide, particularly in many East Asian countries where educational performance is strongly emphasized and outdoor activities are limited [6]. Hong Kong is no exception, and a recent study showed that more than a quarter of children aged 6–8 years were myopic [7,8]. A recent epidemiological study conducted in Japan has shown that the prevalence of myopia is estimated to be 76.5% and 94.9% for schoolchildren aged 6–11 and 12–14 years, respectively [9]. Myopia develops when there is a failure of emmetropization [10,11,12,13,14]. High demands in near work, close working distance, and lack of outdoor activities are known risk factors for myopia development and progression [15,16,17]. This vulnerability to environmental stress necessitates further attention in exploring the consequences of time spent on smart devices, such as smartphones and tablets, for school-aged children’s learning and overall developmental health. People who are myopic have impaired ability to see objects clear at distance, thereby potentially affecting their academic and job performance as well as career choices. Although clinical interventions that can effectively slow myopia progression, such as low dose atropine [18] and orthokeratology [19] are currently available, myopia progression intervention measures should also be implemented through early detection and early childhood education.

Current technological advances have made access to digital devices common among all population groups worldwide. The digitization trend has brought about new habits and demands in lifestyle ergonomics, introducing new public health burdens such as vision-related complications [20]. Schoolchildren are most vulnerable to visual influences, resulting in the onset of subjective symptoms such as visual fatigue, headaches, and blurry vision [21]. Nowadays, there is a surge in smartphone ownership, whereby it is very common to see children as young as two years old using smartphones as their new “electronic pacifier” [22]. The effect of time spent on smart devices among school-aged children has been explored in various studies, most reporting a need for interventional strategies to reduce their usage [23,24]. In Hong Kong, there is an increasing trend in which smart device usage affects sleep patterns among adolescents [25]. A number of studies also indicated that refractive error can be altered by sleep patterns [26,27]. A recent systemic review revealed that screen time was not associated with the prevalence and incidence of myopia [28], nor was it found to be associated with smartphone usage time in a cross-sectional study [29]. However, whether there is any long-term visual impact after smart device usage is yet to be determined [30,31].

Because of the popularity of smartphone use with school children and the introduction of electronically assisted teaching in schools, smart device usage is unavoidable, and appropriate guidelines ought to be developed. Therefore, it is important to explore its consequences for vision so that appropriate education can be provided to end-users to minimize this stress. In this study, we examined the association between smart device usage habits of primary and secondary school-aged children and their 1-year changes in refractive error.

## 2. Materials and Methods

### 2.1. Participants

Participants in this study were recruited in 11 primary schools and 4 secondary schools in Hong Kong. All students from participating schools studying P3–P5 (equivalent to Grades 3 to 5 in the US education system) and S1–S3 (equivalent to Grades 7 to 9 in the US education system) between ages 8 and 14 years were invited, and 1978 participants (response rate 60%) were recruited. They were invited to attend a health examination and completed a self-report questionnaire at baseline and 1-year follow-up. The primary parental caregivers of all participants were also invited to complete a self-report questionnaire at baseline and 1-year follow-up. Written consent was obtained from all participants at baseline. As all participants were under 18 years old, written parental consent was also obtained prior to their participation in this study. This study was approved by the Human Subjects Ethics Sub-committee of Hong Kong Polytechnic University (Reference number HSEARS20151121001). All study procedures followed the guidelines of the Declaration of Helsinki. The details of the study can be found in https://rfs1.fhb.gov.hk/app/fundedsearch/projectdetail.xhtml?id=1958.

### 2.2. Data Collection

#### 2.2.1. Self-Administered Questionnaire

Time spent on smartphones and tablets per day was self-reported, and participants reported the time spent on school days and holidays separately. The daily time spent was calculated as the weighted sum of the time spent on school days and holidays. Time spent on moderate-to-vigorous physical activity (MVPA) was measured using the Global Physical Activity Questionnaire (GPAQ) [32].

#### 2.2.2. Health Examination

Non-cycloplegic refraction was measured by an open-field autorefractor (Grand Seiko WAM-5500). Each eye was measured three times and the average spherical and cylindrical powers were analyzed. SER was computed as the sum of sphere power and half of the cylinder power. Axial length was measured with a Carl Zeiss IOL Master 500 Optical Biometer, with each eye measured five times and averaged.

#### 2.2.3. BMI

The height (to the nearest 0.1 cm) and weight (to the nearest 0.1 kg) of the participants were measured by trained research assistants using a SECA 213 portable stadiometer and Tanita BMI body fat analyzer BC-541N. BMI was computed as weight (kg)/(height [m])^2^.

#### 2.2.4. Caregiver Questionnaire

The primary parental caregiver of each participant (decided among the caregivers) was invited to complete a self-administered questionnaire that collected basic demographic and socio-economic characteristics of the participants (type of accommodation, primary caregiver’s level of education, and monthly household income).

### 2.3. Data Analysis

The sample was weighted according to the 2016 Population By-census to increase its representativeness. Since the 1-year change in the spherical equivalent refraction (SER) of right eyes and left eyes showed only moderate correlation (rho = 0.53), we used multivariate regressions to examine the association between smart device usage and SER instead of analyzing the SER of one eye (mostly the right eye in the literature). Pillai’s trace test was used to examine the overall significance of smart device usage and SER. The SER of right and left eyes were the dependent variables, and numerous possible confounders including age, sex, baseline SER, parents’ short-sightedness, BMI, time spent on MVPA, and caregiver-reported socio-economic status were controlled. The working distance of using smartphones and tablets was not controlled due to its low correlation with SER (right eye: −0.04, left eye: −0.06). Missing data in the confounders were imputed using multiple imputations. The fully conditional specification method was used to impute the missing data, and the average results performed with 10 imputed datasets were reported. All data analysis was performed using SPSS 25.0. (IBM Corp., Armonk, NY, USA) *p*-values of <0.05 were considered significant.

## 3. Results

Table 1 shows the participants’ characteristics. The sample was gender-balanced and the mean age was 10.87 years (SD 2.00). Approximately half of the participants spent more than 4 h per day on smartphones, and approximately 40% spent at least 1 h per day on tablets. The average SERs at baseline were −1.69 D (right eye) and −1.64 D (left eye). At the 1-year follow-up, they progressed to −1.90 D and −1.84 D, respectively.

Time spent on smartphones at baseline was negatively and significantly associated with the SER of both eyes at baseline and 1-year follow-up (Table 2, both *p*s < 0.001). Time spent on tablets was insignificantly associated with baseline SER but significantly associated with SER at 1-year follow-up (both *p*s < 0.05), with participants who spent 2–3 h per day on tablets having had the most negative SER (−2.30 D for the right eye and −2.21 D for the left eye), that is, having the most negative refractive error. Time spent on both smartphones and tablets at baseline was insignificantly associated with changes in the SER of both eyes.

Unadjusted and adjusted analyses of the association between smart device usage and changes in SER showed similar results (Table 3 and Table 4). *p*-values from Pillai’s trace were insignificant. When the time spent on smartphones was 2–3, 3–4, and 4+ hours per day, the adjusted means of the 1-year change in SER ranged from −0.20 to −0.26 D and were all significantly different from zero. When the time spent on tablets was 0–1, 2–3, and 3–4 h per day, the adjusted means of the 1-year change in SER ranged from −0.18 to −0.23 D, and were all significantly different from zero.

From the above results, 2 h per day of smart device usage appeared to be a cutoff for SER change, and this cutoff was thus used in the following analysis. Table 5 shows the interactive association of smartphone and tablet usage on the 1-year change in SER. The high smartphone usage (≥2 h per day) and low tablet usage (<2 h per day) subgroup was the group with the highest negative refractive error at both baseline and 1-year follow-up (both *p*s < 0.001). Multivariate regression was performed to examine the interactive effect of smartphone and tablet usage on the 1-year change in SER while controlling for potential confounders (Table 6). For right eyes, compared with the reference group (those with <2 h per day on both smartphone and tablet usages), individuals spending ≥2 h per day on smartphone usage and <2 h per day on tablet usage had a significantly larger decrease in SER (1-year change −0.09 vs. −0.25 D, *p* = 0.01), while the level of significance was marginal (1-year change −0.15 vs. −0.28 D, *p* = 0.055) for the left eye. The other two subgroups had insignificant differences in SER compared with those of the reference group.

## 4. Discussion

In this study, we found that smartphone use in young children was associated with a negative shift in refractive error. Children who spent more time (≥2 h per day) on smartphones, but less time (<2 h per day) on tablets showed greater negative shift in refractive error than those who spent more time on both devices. These results suggested that prolonged smartphone usage may present a higher risk of myopia progression than tablet usage. The study results suggested that children and adolescents should spend at most 2 h per day on both smartphones and tablets to reduce myopia progression. It is believed that tablet usage has less impact on SER because of the difference in posture when one uses tablets and smartphones. Studies have shown that people tend to place tablets further away than smartphones during use [33], the convergence demand during tablet use is less than that of smartphone use [34], and this prolonged accommodative convergence may contribute to myopia progression [35].

This study explored a dilemma in the current education system. While smart devices have been widely used to augment learning through online teaching, whether they pose long-term visual repercussions such as myopia progression in school children is yet to be confirmed. Given this uncertainty regarding the prevalence of myopia, when increased smart device usage is necessary, a monitoring model should be applied until further evidence clearly indicates the long-term effects of electronic screen time on children’s myopia development. Our results showed that school-children who spent more time on smart devices had a higher magnitude of myopia, at both the baseline and 1-year follow-up measurements. This finding is consistent with a recent cross-sectional study conducted in urban areas of Tianjin [36]. On the other hand, the myopia progression, as determined by SER changes over a 1-year period, did not differ significantly among groups with different smartphone/tablet usage. Nonetheless, we observed that school-aged children who reported using smart devices for two hours or more per day tended to have a greater increase in myopia change within a 1-year follow-up period. This discrepancy could be caused by the limited duration of the study, as the annual changes in refractive error in Hong Kong schoolchildren are estimated to be 0.5D [37], rendering the detection of subtle differences challenging. In addition, the negative correlation between smartphone/tablet usage and SER at the baseline visit may suggest a potential causal relationship, as a child might have already been using smartphones/tablets for years before the baseline measurement in this study. Although myopia progression was evident in all groups at the 1-year follow-up, its progression parallels in significance to the duration of usage. Further studies with more frequent measurements are required to confirm this finding by monitoring screen time and changes in SER for a longer period.

Consistent with previous findings, this study factored in potentially extraneous factors that could lead to an increase in myopia, including the parental history of myopia [8,38]. Our sampling included the core public schools representing the typical Hong Kong education experience, which further highlights the need for myopia progression monitoring. One limitation of our study was that cycloplegic agents were not used due to ethical issues, as the data were collected on normal school days and cycloplegic agent installation would interrupt the subjects’ daily school activities. Lack of cycloplegic agents may potentially affect the results of the autorefraction, as the subjects may have a tendency to accommodate, resulting in a more negative SER [39]. However, our data showed that SERs were strongly correlated with the axial length measurements (baseline *r* = 0.74, *p* < 0.001), and hence, we expected that the measured SER should also be strongly correlated with the actual SER. Other limitations include a more objective measure of screen time (e.g., screen time app) usage that tracks students’ usage over the year, which may be required to provide more accurate results on the usage of devices rather than questionnaire-based data. With that, screen time usage over the entire experimental period can be monitored more precisely. Finally, the moderately-correlated 1-year change in SER of right and left eyes might be contributed by the position of the smart device during use, but we did not collect data on the hand they commonly used for holding the smart devices so our hypothesis could not be tested.

## 5. Conclusions

In summary, based on the 1-year data, we suggest that additional consideration may be needed when introducing new forms of learning using smart devices. Our data indicated the prevalence of their habitual use and the potentially detrimental consequences for eye health. Our results will allow public health practitioners to steer strategic evaluations of the need for potential intervention strategies for children regarding screen-time control. Future research should also be directed towards exploring school health service models on how we could enhance the delivery of eye screening and/or examination to students, as well as educate them about the potential impact of screen time on eye health.

## Figures and Tables

**Table 1 ijerph-17-08923-t001:** Descriptive statistics (*n* = 1597).

Variable	Mean	SD
Age (years)	10.87	2.00
Spherical equivalent refraction (D)		
Right eye, baseline	−1.69	2.14
Right eye, 1-year follow-up	−1.90	2.20
Right eye, change	−0.21	1.01
Left eye, baseline	−1.64	2.12
Left eye, 1-year follow-up	−1.84	2.18
Left eye, change	−0.20	1.00
BMI (kg/m^2^)	18.55	3.56
MVPA (h/wk)	11.62	14.53
**Variable**	**Frequency**	**Percentage**
Gender		
Male	797	49.9
Female	800	50.1
Smartphone usage (h/day)		
0–1	295	18.5
1–2	184	11.5
2–3	145	9.1
3–4	167	10.5
4+	806	50.5
Tablet usage (h/day)		
0–1	990	62.0
1–2	188	11.8
2–3	116	7.3
3–4	73	4.6
4+	230	14.4
Parents’ short-sightedness		
Both	298	19.0
Only father	302	19.2
Only mother	234	14.9
Neither	530	33.7
Do not know	208	13.2
Type of accommodations		
Public housing	1021	67.6
Home-ownership scheme	193	12.8
Private housing	296	19.6
Primary caregiver’s level of education		
No formal education	17	1.1
Primary	182	12.3
Secondary	1127	76.0
Tertiary or above	157	10.6
Monthly household income		
0–9999	219	14.9
10,000–19,999	589	40.1
20,000–29,999	343	23.3
30,000–39,999	163	11.1
40,000–49,999	76	5.2
50,000+	79	5.4

BMI: Body Mass Index, MVPA: moderate-to-vigorous physical activity.

**Table 2 ijerph-17-08923-t002:** Univariate association between smartphone and tablet usage (hours per day) on spherical equivalent refraction (SER, D).

Smartphone	Usage (h/day)	Baseline	1-Year Follow-Up	Change
	Right eye			
	0–1	−1.21 (2.10)	−1.38 (2.06)	−0.17 (1.17)
	1–2	−1.55 (2.00)	−1.73 (1.94)	−0.17 (1.07)
	2–3	−1.75 (2.29)	−2.05 (2.41)	−0.30 (0.70)
	3–4	−1.78 (2.27)	−1.97 (2.46)	−0.19 (0.98)
	4+	−1.87 (2.10)	−2.09 (2.18)	−0.22 (0.98)
	*p*-value	<0.001	<0.001	0.72
	Left eye			
	0–1	−1.16 (2.06)	−1.38 (2.04)	−0.22 (1.20)
	1–2	−1.56 (1.97)	−1.75 (2.03)	−0.19 (1.04)
	2–3	−1.76 (2.24)	−2.02 (2.36)	−0.26 (0.70)
	3–4	−1.63 (2.28)	−1.88 (2.28)	−0.25 (0.84)
	4+	−1.81 (2.10)	−1.99 (2.19)	−0.18 (0.99)
	*p*-value	<0.001	0.001	0.84
**Tablet**	**Usage (h/day)**	**Baseline**	**1-Year Follow-Up**	**Change**
	Right eye			
	0–1	−1.72 (2.18)	−1.96 (2.27)	−0.24 (1.01)
	1–2	−1.53 (2.00)	−1.62 (1.88)	−0.09 (1.14)
	2–3	−2.06 (2.39)	−2.30 (2.48)	−0.24 (0.90)
	3–4	−1.33 (2.06)	−1.58 (1.95)	−0.25 (0.87)
	4+	−1.62 (1.93)	−1.78 (1.99)	−0.16 (0.96)
	*p*-value	0.13	0.04	0.37
	Left eye			
	0–1	−1.70 (2.18)	−1.92 (2.26)	−0.23 (1.05)
	1–2	−1.42 (1.98)	−1.53 (1.88)	−0.11 (0.95)
	2–3	−1.93 (2.26)	−2.21 (2.38)	−0.27 (0.81)
	3–4	−1.28 (2.00)	−1.48 (1.87)	−0.20 (1.05)
	4+	−1.51 (1.94)	−1.66 (1.98)	−0.15 (0.92)
	*p*-value	0.10	0.02	0.48

**Table 3 ijerph-17-08923-t003:** Multivariate regression of smartphone usage (hours per day) on 1-year change in spherical equivalent refraction (SER), imputed data.

	Model 1		Model 2		Model 3	
Usage (h/day)	Mean (95% CI)	*p*-Value (Comparison with 0–1)	Mean (95% CI)	*p*-Value (Comparison with 0–1)	Mean (95% CI)	*p*-Value (Comparison with 0–1)
Right eye						
0–1	−0.12 (−0.24, −0.01)	Ref	−0.11 (−0.24, 0.003)	Ref	−0.11 (−0.26, 0.04)	Ref
1–2	−0.13 (−0.27, 0.02)	0.97	−0.14 (−0.29, 0.01)	0.79	−0.11 (−0.28, 0.07)	0.83
2–3	−0.28 (−0.44, −0.13)	0.10	−0.29 (−0.46, −0.13)	0.12	−0.25 (−0.44, −0.06)	0.09
3–4	−0.18 (−0.33, −0.04)	0.53	−0.21 (−0.36, −0.06)	0.71	−0.20 (−0.38, −0.01)	0.30
4+	−0.25 (−0.32, −0.18)	0.08	−0.27 (−0.34, −0.19)	0.07	−0.23 (−0.37, −0.10)	0.04
Left eye						
0–1	−0.60 (−0.28, −0.04)	Ref	−0.14 (−0.26, −0.02)	Ref	−0.16 (−0.31, −0.01)	Ref
1–2	−0.16 (−0.30, −0.01)	0.97	−0.15 (−0.30, −0.01)	0.90	−0.17 (−0.34, 0.01)	0.93
2–3	−0.26 (−0.41, −0.10)	0.33	−0.26 (−0.42, −0.10)	0.12	−0.26 (−0.45, −0.07)	0.27
3–4	−0.19 (−0.34, −0.05)	0.74	−0.23 (−0.38, −0.08)	0.71	−0.25 (−0.43, −0.07)	0.38
4+	−0.22 (−0.29, −0.15)	0.42	−0.23 (−0.30, −0.15)	0.07	−0.24 (−0.37, −0.11)	0.28
Pillai’s trace *p*-value	0.68		0.59		0.59	

Model 1 adjusted for age, sex, and baseline SER; Model 2 adjusted for age, sex, baseline SER, parents’ short-sightedness, BMI, and time spent on moderate-to-vigorous physical activity; Model 3 adjusted for age, sex, baseline SER, parents’ short-sightedness, BMI, time spent on moderate-to-vigorous physical activity, and caregiver-reported socio-economic status.

**Table 4 ijerph-17-08923-t004:** Multivariate regression of tablet usage (hours per day) on 1-year change in spherical equivalent refraction (SER), imputed data.

	Model 1		Model 2		Model 3	
Usage (h/day)	Mean (95% CI)	*p*-Value (Comparison with 0–1)	Mean (95% CI)	*p*-Value (Comparison with 0–1)	Mean (95% CI)	*p*-Value (Comparison with 0–1)
Right eye						
0–1	−0.24 (−0.30, −0.18)	Ref	−0.25 (−0.32, −0.19)	Ref	−0.22 (−0.34, −0.09)	Ref
1–2	−0.09 (−0.23, 0.05)	0.053	−0.09 (−0.23, 0.05)	0.04	−0.07 (−0.24, 0.10)	0.04
2–3	−0.25 (−0.42, −0.07)	0.94	−0.28 (−0.46, −0.10)	0.80	−0.22 (−0.42, −0.02)	0.80
3–4	−0.21 (−0.44, 0.01)	0.84	−0.22 (−0.44, 0.01)	0.76	−0.18 (−0.43, 0.06)	0.74
4+	−0.16 (−0.28, −0.03)	0.26	−0.17 (−0.30, −0.04)	0.27	−0.14 (−0.31, 0.03)	0.28
Left eye						
0–1	−0.24 (−0.30, −0.18)	Ref	−0.24 (−0.30, −0.18)	Ref	−0.23 (−0.36, −0.10)	Ref
1–2	−0.08 (−0.22, 0.06)	0.08	−0.07 (−0.22, 0.07)	0.03	−0.10 (−0.27, 0.07)	0.04
2–3	−0.26 (−0.43, −0.08)	0.09	−0.28 (−0.46, −0.10)	0.71	−0.26 (−0.46, −0.05)	0.80
3–4	−0.18 (−0.40, 0.04)	0.12	−0.16 (−0.39, 0.06)	0.52	−0.18 (−0.42, 0.06)	0.49
4+	−0.13 (−0.25, −0.03)	0.07	−0.14 (−0.27, −0.01)	0.16	−0.15 (−0.32, 0.02)	0.14
Pillai’s trace *p*-value	0.59		0.52		0.52	

Model 1 adjusted for age, sex, and baseline SER; Model 2 adjusted for age, sex, baseline SER, parents’ short-sightedness, BMI, and time spent on moderate-to-vigorous physical activity; Model 3 adjusted for age, sex, baseline SER, parents’ short-sightedness, BMI, time spent on moderate-to-vigorous physical activity, and caregiver-reported socio-economic status.

**Table 5 ijerph-17-08923-t005:** Interactive association between smartphone and tablet usage (hours per day) on spherical equivalent refraction (SER, D).

	Smartphone Usage (h/day)	Tablet Usage (h/day)	Baseline	1-Year Follow-Up	Change
Right eye					
	<2	<2	−1.29 (2.02)	−1.46 (1.98)	−0.17 (1.18)
	≥2	<2	−1.88 (2.19)	−2.12 (2.30)	−0.24 (0.96)
	<2	≥2	−1.54 (2.22)	−1.72 (2.16)	−0.18 (0.90)
	≥2	≥2	−1.73 (2.07)	−1.94 (2.14)	−0.20 (0.93)
*p*-value			<0.001	<0.001	0.68
Left eye					
	<2	<2	−1.27 (2.02)	−1.47 (2.04)	−0.20 (1.20)
	≥2	<2	−1.84 (2.18)	−2.05 (2.27)	−0.21 (0.94)
	<2	≥2	−1.48 (2.08)	−1.74 (2.05)	−0.27 (0.89)
	≥2	≥2	−1.62 (2.04)	−1.79 (2.10)	−0.17 (0.92)
*p*-value			<0.001	<0.001	0.86

**Table 6 ijerph-17-08923-t006:** Multivariate regression of the interactive effect of smartphone and tablet usage (<2 h per day vs. ≥2 h per day) on 1-year change in spherical equivalent refraction (SER), imputed data.

	Smartphone Usage (h/day)	Tablet Usage (h/day)	Model 1		Model 2		Model 3	
			Mean (95% CI)	*p*-Value (Comparison with <2 Hours per Day on Both Smartphone and Tablet Usages)	Mean (95% CI)	*p*-Value (Comparison with 0–1)	Mean (95% CI)	*p*-Value (Comparison with <2 Hours per Day on Both Smartphone and Tablet Usages)
Right eye								
	<2	<2	−0.10 (−0.21, −0.01)	Ref	−0.12 (−0.22, −0.01)	Ref	−0.09 (−0.24, 0.05)	Ref
	≥2	<2	−0.27 (−0.34, −0.20)	0.02	−0.28 (−0.35, −0.21)	0.02	−0.25 (−0.38, −0.13)	0.01
	<2	≥2	−0.15 (−0.35, 0.05)	0.73	−0.16 (−0.36, 0.05)	0.73	−0.13 (−0.36, 0.09)	0.71
	≥2	≥2	−0.21 (−0.32, −0.11)	0.18	−0.22 (−0.33, −0.11)	0.17	−0.20 (−0.34, −0.05)	0.17
Left eye								
	<2	<2	−0.13 (−0.23, −0.03)	Ref	−0.13 (−0.23, −0.02)	Ref	−0.15 (−0.29, −0.003)	Ref
	≥2	<2	−0.25 (−0.32, −0.18)	0.07	−0.26 (−0.33, −0.18)	0.051	−0.28 (−0.40, −0.15)	0.055
	<2	≥2	−0.22 (−0.42, −0.01)	0.11	−0.21 (−0.42, −0.01)	0.45	−0.22 (−0.44, 0.0005)	0.52
	≥2	≥2	−0.17 (−0.27, −0.06)	0.08	−0.17 (−0.28, −0.06)	0.57	−0.18 (−0.33, −0.04)	0.64
Pillai’s trace *p*-value			0.27		0.002		0.25	

Model 1 adjusted for age, sex, and baseline SER; Model 2 adjusted for age, sex, baseline SER, parents’ short-sightedness, BMI, and time spent on moderate-to-vigorous physical activity; Model 3 adjusted for age, sex, baseline SER, parents’ short-sightedness, BMI, time spent on moderate-to-vigorous physical activity, and caregiver-reported socio-economic status.

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
