# Peer review of "Association between Time Spent on Smart Devices and Change in Refractive Error: A 1-Year Prospective Observational Study among Hong Kong Children and Adolescents"

_ijerph, 2020, doi:10.3390/ijerph17238923_

Round 1

Reviewer 1 Report

The manuscript is well written, well organized with proper statistical analysis. I have the following minor points for the authors to address:

  1. Line 92: It is better to use “moderate-to-vigorous physical activity (MVPA)” here instead of “MVPA”, since it is the first appear in the text (or alternatively, can do it in the abstract line 25). And, line 112 should use MVPA.
  2. Line 148 to line 151: Missing Models 1, 2 & 3.
  3. Since working distance is an important factor for myopia development, it would be a plus if the authors can also report the working distance of using smartphones and tablets other than only report the time spent. If the questionnaire did not include questions of working distance, the authors can discuss it as study limitation or need further studies, etc.

Author Response

The point-by-point response is attached.

Reviewer 2 Report

Chi-wai et al. report the results of a novel prospective analysis of tablet and smart-phone usage on myopic progression. There is an abundance of similar evidence in the current literature (see https://doi.org/10.1111/opo.12657 for a review), but most studies are retrospective or cross-sectional in nature and deal with prevalence or incidence of myopia from digital device use. The current study significantly adds to the current body of knowledge by comparing the effects of smart-phone vs. tablet use on myopic progression.  The authors adequately (perhaps even "over-") analyze the relevant effects, but all of the analyses appear appropriate and sound. The analysis was very detailed, and it is possible that the univariate ad-hoc analysis would benefit from post-hoc corrections. However, no conclusions are over-stated. I enjoyed this paper, and I thank the editorial staff and authors for the first look.

With that said, I have some minor suggestions for the journal staff and authors:

MAJOR: N/A

MINOR:

1.  Lines 86-87: According to the Declaration of Helsinki, study must be registered in publicly accessible storehouse (Article 16). Please remove statement regarding "Declaration of Helsinki..." (the current reference number is adequate) or provide link to where study is registered. 

2. Lines 88-205: Please format headings to journal specifications.

3. Line 100: BMI = "weight (kg)/height (m)^2" is awkward. Suggest "weight (kg)/(height [m])^2

4. Lines 108-109: Suggest using dashes "--" instead of commas to separate this clause.

5. Line 113: I suggest using "missing data" instead of "missing confounders"  The method is to average (or impute) data and fill in gaps, not add "confounds".

6. Table 1: This table is simply ugly. It takes a lot of effort to sort out the subheadings, etc. Please revise, using single-spacing in sub-headings, etc.

7. Table 4: Why did the authors report all MANOVA stats? I suggest choosing one (e.g., Pillai's trace) to report and remove others.

Thanks you, again, for the opportunity to review.

Author Response

Comment 1:  Lines 86-87: According to the Declaration of Helsinki, study must be registered in publicly accessible storehouse (Article 16). Please remove statement regarding "Declaration of Helsinki..." (the current reference number is adequate) or provide link to where study is registered. 

Response 1: The details of the study could be found in the website of the funder, Food and Health Bureau, Hong Kong SAR Government (https://rfs1.fhb.gov.hk/app/fundedsearch/projectdetail.xhtml?id=1958). We have provided this link in the Methods section (lines 91 to 92).

Comment 2: Lines 88-205: Please format headings to journal specifications.

Response 2: We have now format the headings according to journal specifications.

Comment 3: Line 100: BMI = "weight (kg)/height (m)^2" is awkward. Suggest "weight (kg)/(height [m])^2

Response 3: Done.

Comment 4: Lines 108-109: Suggest using dashes "--" instead of commas to separate this clause.

Response 4: We have revised the sentence as “Since the 1-year change in the SER of right eyes and left eyes showed only moderate correlation (rho=0.53), this study, we used multivariate regressions to examine the association between smart device usage and SER instead of analyzing the SER of one eye (mostly the right eye in the literature).”.(lines 116 to 118).

Comment 5: Line 113: I suggest using "missing data" instead of "missing confounders"  The method is to average (or impute) data and fill in gaps, not add "confounds".

Response 5: We have revised the sentence as “Missing data in the confounders were…” (line 125).

Comment 6: Table 1: This table is simply ugly. It takes a lot of effort to sort out the subheadings, etc. Please revise, using single-spacing in sub-headings, etc.

Response 6: We have revised Table 1 with single-spacing in sub-headings.

Comment 7: Table 4: Why did the authors report all MANOVA stats? I suggest choosing one (e.g., Pillai's trace) to report and remove others.

Thanks you, again, for the opportunity to review.

Response 7: We now only retained the results of Pillai’s trace test in Tables 3, 4, and 6. Relevant descriptions in the main text about the Methods (lines 120 to 121) and Results (lines 151 to 152) have also been revised accordingly.

Reviewer 3 Report

This is an interesting topic and important clinically.  A few questions arise:  

With regard to the differences found between right and left eyes, was ocular dominance tested?  

Were any of the subjects wearing their spectacle correction at any time during the observational period?  

What was the position of the smart device relative to the eyes?  (The amount of accommodation would be expected to be the same for each eye, but I am looking for an explanation of the difference between the two eyes.)

The sentence on lines 68-69 is confusing.

Author Response

This is an interesting topic and important clinically.  A few questions arise:  

Comment 1: With regard to the differences found between right and left eyes, was ocular dominance tested?  

Response 1: Ocular dominance was not tested in this study, therefore we were unable to test the differences found between the right and left eyes.

Comment 2: Were any of the subjects wearing their spectacle correction at any time during the observational period?  

Response 2: We assumed that all subjects wore their habitual spectacle corrections during the observational period.

Comment 3: What was the position of the smart device relative to the eyes?  (The amount of accommodation would be expected to be the same for each eye, but I am looking for an explanation of the difference between the two eyes.)

Response 3: The average working distance was 19.8 cm (SD 14.3). We did not collect the data about the hand used to hold the smart device. We have acknowledged this as a limitation as “Finally, the moderately-correlated 1-year change in SER of right and left eyes might be contributed by the position of smart device during use, but we did not collect the hand they commonly used for holding the smart devices so that our hypothesis could not be tested.” (lines 246 to 249).

Comment 4: The sentence on lines 68-69 is confusing.

Response 4: We have revised this sentence as “However, it remains to be determined whether there is any long-term visual impact after smart device usage is yet to be determined  [30, 31].” (line 70).

Reviewer 4 Report

There is a crucial problem in this study.  Non-cycloplegic auto-refraction does not provide an accurate reflection of myopic progression.  Despite an intriguing question, there is a substantial flaw in the design.

Author Response

Comment 1: There is a crucial problem in this study.  Non-cycloplegic auto-refraction does not provide an accurate reflection of myopic progression.  Despite an intriguing question, there is a substantial flaw in the design.

Response 1: While we agree that cycloplegic auto-refraction is the best approach, cycloplegic agents causes the pupil dilatation that can last for the whole school day. Therefore, it was extremely difficult to obtain school approval to use cycloplegic agent.

Given the above reason, non-cycloplegic auto-refraction has been widely used in large-scale school-based research. Examples that relied on non-cycloplegic auto-refraction published in 2020 include:

Fu, A., et al. (2020) Prevalence of myopia among disadvantaged Australian schoolchildren: A 5-year cross-sectional study. PLoS ONE 15(8): e0238122.

Hansen, M. H., et al. (2020) Low physical activity and higher use of screen devices are associated with myopia at the age of 16-17 years in the CCC2000 Eye Study. Acta Ophthalmologica 98, 315-321.

Gessesse, S. A., et al. (2020) Prevalence of myopia among secondary school students in Welkite town: South-Western Ethiopia BMC Ophthalmology 20:176.

Junghans, B. M., et al. (2020) Unexpectedly high prevalence of asthenopia in Australian school children identified by the CISS survey tool. BMC Ophthalmology 20:408.

Wang, J., et al. (2020) School-based epidemiology study of myopia in Tianjin, China. International Ophthalmology, 40, 2213-2222.

Yang, L., et al. (2020) Thirty-five-year trend in the prevalence of refractive error in Austrian conscripts based on 1.5 million participants. British Journal of Ophthalmology, 104, 1338-1344.

To evaluate the reliability of our non-cycloplegic refractive outcome, we provided its correlation with axial length measurement (r=0.74) so we believe that our results have a high level of validity.